Proteomic profiling analysis of postmenopausal osteoporosis and osteopenia identifies potential proteins associated with low bone mineral density

Huang Dageng 1
Wang Yangyang 2
http://orcid.org/0000-0003-2801-743X Lv Jing 3
Yan Yuzhu 3
Hu Ya 4
Liu Cuicui 1
Zhang Feng 5
Wang Jihan 3 513837742@qq.com
Hao Dingjun 1 haodingjun@126.com
1 Department of Spine Surgery, Honghui Hospital, Xi’an Jiaotong University , Xi’an , China
2 School of Electronics and Information, Northwestern Polytechnical University , Xi’an , China
3 Clinical Laboratory of Honghui Hospital, Xi’an Jiaotong University , Xi’an , China
4 Department of Physiology, Hunan Polytechnic of Environment and Biology , Hengyang , China
5 School of Public Health, Health Science Center, Xi’an Jiaotong University , Xi’an , China
Uversky Vladimir
Electronic publication date: 2020 Apr 14
Publication date: 2020
Volume: 8
Electronic Location ID: e9009
Received 2019 Dec 23; Accepted 2020 Mar 27
Copyright: © 2020 Huang et al.
Copyright year: 2020
Copyright holder: Huang et al.
License: This is an open access article distributed under the terms of the Creative Commons Attribution License, which permits unrestricted use, distribution, reproduction and adaptation in any medium and for any purpose provided that it is properly attributed. For attribution, the original author(s), title, publication source (PeerJ) and either DOI or URL of the article must be cited.
License URL: https://creativecommons.org/licenses/by/4.0/

Keywords: Postmenopausal osteoporosis, Bone mineral density, Proteomics, TMT, PRM, Diagnostic biomarkers, Therapeutic targets

Funding: National Natural Science Foundation of China 81702067 Natural Science Foundation of Shaanxi Province 2017JQ8041 and 2019JQ-978 Young Talents Supporting Project of Xi’an Association for Science and Technology (2019) This work was supported by the National Natural Science Foundation of China (No. 81702067), the Natural Science Foundation of Shaanxi Province (No. 2017JQ8041 and 2019JQ-978), and the Young Talents Supporting Project of Xi’an Association for Science and Technology (2019). The funders had no role in study design, data collection and analysis, decision to publish, or preparation of the manuscript.

==============================
Postmenopausal osteoporosis (PMOP) is a major global public health concern and older women are more susceptible to experiencing fragility fractures. Our study investigated the associations between circulating proteins with bone mineral density (BMD) in postmenopausal women with or without low BMD (osteoporosis and osteopenia) using a tandem mass tag (TMT) labeling proteomic experiment and parallel reaction monitoring testing. Across all plasma samples, we quantitatively measured 1,092 proteins, and the OP and normal control (NC) samples were differentiated by principal component analysis and a partial least squares-discrimination analysis model based on the protein profiling data. The differentially abundant proteins between the low BMD and NC samples mostly exhibited binding, molecular function regulator, transporter and molecular transducer activity, and were involved in metabolic and cellular processes, stimulus response, biological regulation, immune system processes and so forth. TMT analysis and RRM validation indicated that the expression of protein Lysozyme C (P61626) was negatively related to BMD, while the expression of proteins Glucosidase (A0A024R592) and Protein disulfideisomerase A5 (Q14554) was positively related to BMD values. Collectively, our results suggest that postmenopausal women with low BMD have a different proteomic profile or signature. Protein alterations may play an important role in the pathogenesis of PMOP, and they may act as novel biomarkers and targets of therapeutic agents for this disease.

Introduction

Osteoporosis (OP) is a common metabolic bone disorder characterized by the microarchitectural deterioration of bone tissues, low bone mineral density (BMD), and increased fracture risk. Menopause and aging are two critical factors that are directly related to OP onset (Cheishvili et al., 2018). Postmenopausal osteoporosis (PMOP) is a major global public health concern that frequently presents in postmenopausal women due to estrogen deficiency and the continuous calcium loss that occurs with aging (Thaung Zaw, Howe & Wong, 2018). Osteoporotic fracture (OF) caused by OP is a major hazard and is one of the main causes of disability and death in older adults (Qaseem et al., 2017). It is estimated that there will be about 4.83 million OF cases in 2,035 and about 5.99 million cases in 2050 (Si et al., 2015). A woman’s risk of developing OF during her lifetime (40%) is greater than the risk for ovarian cancer, breast cancer, and endometrial cancer combined (Lin et al., 2015). In response to this, studies exploring novel biomarkers and the bone metabolism-related mechanisms underlying PMOP that aid in its early diagnosis are urgently needed and relevant.

At present, there is no ideal method to measure or evaluate bone strength directly. Clinically, the BMD value measured by dual-energy X-ray absorptiometry (DXA) is used to diagnose OP and predict the risk of OF (Dimai, 2017). Bone metabolism is a dynamic process and current research has not yet clarified the pathogenesis of osteopenia (ON) or OP; it is generally accepted that the imbalance between osteoblast and osteoclast metabolism plays a key role in the pathogenesis of OP (Lewiecki, 2011). When bone metabolism changes, the levels of serum bone turnover markers (BTMs) can change in as early as 3 days, while BMD can change after 1 month (McClung et al., 2006). Further, high-frequency detection of BMD is unrealistic, irrespective of whether the bone density value has changed. In clinical practice, the DXA–BMD measurement is usually performed every 6 months or annually for high-risk groups or for those who receive medication to monitor the natural course of the disease and evaluate the efficacy of various drugs for OP. Therefore, some biomarker levels may exhibit changes in the early stages of abnormal bone metabolism, particularly when the BMD value may not meet the criteria for ON or OP, and the body fluid testing for metabolic molecules is more convenient and implementable for BMD measurement. However, there are many meaningful studies such as serum biomarkers may be associated with high bone turnover and BMD in postmenopausal women (Bhattacharyya et al., 2008) and proteomic analysis reveals Vitamin D-Binding Protein as a potential biomarker for low BMD in Mexican postmenopausal women (Martínez-Aguilar et al., 2019). They are outstanding inspiration for our study.

When compared with the findings of a genome and transcriptome study, proteomics and metabolomics appears to be closer to the phenotype, so it is more suitable for mechanistic studies, disease typing, and biomarker discovery (Jimenez-Munguia et al., 2018; Li et al., 2019; Li & Shui, 2019; Wilmanski et al., 2019). Previous human studies have identified a few candidate metabolites for BMD alteration (Miyamoto et al., 2018; You et al., 2014), while more research about proteome studies is needed to better understand the molecular changes associated with low BMD in OP and ON, especially in postmenopausal women who are more susceptible to experiencing fragility fractures. Herein, we performed mass spectrometry (MS)-based proteomic experiments with tandem mass tag (TMT) labeling and further validated a few candidate differentially abundant proteins (DAPs) obtained from the MS data by means of parallel reaction monitoring (PRM) analysis in postmenopausal women with normal bone mass, ON and OP. Our aim was to identify novel candidate proteins that are associated with BMD changes in postmenopausal women with a low BMD, while also revealing the mechanism of abnormal bone metabolism and possible diagnostic biomarkers underlying this widespread disease.

Materials and Methods

Study populations and design

For this study, we recruited a total of 54 postmenopausal females, including 18 female patients with primary PMOP, 18 with postmenopausal ON, and 18 with normal bone mass as NCs were recruited from March 2018 to February 2019 in Honghui Hospital, Xi’an Jiaotong University (People’s Republic of China). Patients with secondary OP, including those with Cushing’s syndrome, hyperparathyroidism, hyperthyroidism, steroid abuse, and other malignancies such as chronic liver disease, heart disease and kidney disease were excluded. Drug factors including glucocorticoids, immunosuppressants, heparin, anticonvulsants, anticancer drugs, aluminum-containing antacids, thyroid hormones, etc. were also excluded. In addition, BMD measurements were not available in the following situations: (1) those orally administered with drugs within 2–6 days of measurement that affect image development; (2) when radioisotope inspections were carried out within 3 days; (3) those who cannot lie on the examination bed, or cannot adhere to the 5-min test; (4) those with a severe deformity of the spine or metal implants on the spine. Finally, biochemical detection of serum 25-hydroxyvitamin D (vitamin D), alkaline phosphatase (ALP), procollagen type 1 N-peptide (P1NP) and C-terminal cross-linking telopeptide of type 1 collagen (CTX) concentrations as well as BMD measurements were determined for all participants. Each participant signed an informed consent form, and the project was approved by the Institutional Review Board, Honghui Hospital, Xi’an Jiaotong University (Project No: 2018-22). We collected participants’ information, including their age, age at menopause, height (m) and body mass (kg). Participants’ BMI was calculated as mass/height2. The experimental workflow is presented in Fig. 1.

Figure 1 Experimental workflow.

TMT, tandem mass tag; LC–MS/MS, liquid chromatography–tandem mass spectrometry; DDA, data-dependent acquisition; PRM, parallel reaction monitoring.

BMD measurement

Dual-energy X-ray absorptiometry was used to detect the BMD of the lumbar spine (L1–4) and hip (Discovery Wi, Hologic, Marlborough, MA, USA) in all participants. The scan images were obtained from the detector and are shown in Fig. 2. We collected information pertaining to patients’ BMD values (g/cm2) and T scores. The T-score reference ranges were calculated using data from healthy young Asian women provided by the bone densitometry equipment manufacturer. Participants were categorized as having OP, ON, or normal BMD according to the World Health Organization’s T-score classification: participants with T-scores ≤–2.5 at any site were diagnosed as having OP, –2.5 < T-scores < –1.0 had ON and T-scores ≥ –1.0 were indicative of normal bone mass. The PRM approach has been used for BMD-related DAPs validation.

Figure 2 DXA scan image of the lumbar spine (L1–4) and hip for BMD measurement.

(A and B) Lumbar and hip in normal BMD group. (C and D) Lumbar and hip in osteopenia group. (E and F) Lumbar and hip in osteoporosis group.

Blood sample collection

We collected the fasting venous blood samples from each participant, and the blood samples were centrifuged at 4 °C at 3,000 rpm for 15 min. After the centrifugal, the supernatant from each sample were collected and stored at −80 °C until measurement. In each group, we randomly mixed six plasma samples into one sample for the TMT proteome experiment and PRM validation.

Biochemical detection

Serum 25-hydroxyvitamin D (Vitamin D), alkaline phosphatase (ALP), procollagen type 1 N-peptide (P1NP), and C-terminal cross-linking telopeptide of type 1 collagen (CTX) were detected using the Roche electrochemiluminescence system (Cobas e 601 analyzer, Roche Diagnostics GmbH, Wetzlar, Germany).

Protein extraction and enzymolysis

The plasma pools were depleted of the proteins by using the Multiple Affinity Removal LC Column-Human 14 (Cat. 5188-6560, Agilent, CA, USA) according to the protocol from the manufacturer. The desalination and concentration of the low-abundance components were performed by using a 10 kDa ultrafiltration tube. One volume of SDT (4% SDS, 100 mM Tris/HCl pH 7.6, 0.1 M dithiothreitol) buffer was added, and then boiled for 15 min, then centrifuged for 20 min with 14,000×g. The BCA Protein Assay Kit (Bio-Rad, Hercules, CA, USA) was used for protein quantification, and all the supernatant samples were stored at −80 °C for further measurement.

SDS–PAGE separation

Overall, proteins (20 µg) were mixed by the 5× loading buffer for each sample, respectively, and boiled for 5 min. Proteins from all the samples were separated by a 12.5% sodium dodecyl sulfate (SDS)-polyacrylamide gel electrophoresis (PAGE) gel and the Coomassie Blue R-250 staining system was used for binds visualization.

TMT labeling

A total of 200 μg of proteins for each sample were incorporated into 30 μl SDT buffer (4% SDS, 100 mM DTT, 150 mM Tris-HCl pH 8.0). The detergent, DTT and other low-molecular-weight components were removed using UA buffer (8 M Urea, 150 mM Tris-HCl pH 8.0) by repeated ultrafiltration (Microcon units, 10 kD). Then 100 μl iodoacetamide (100 mM IAA in UA buffer) was added to block reduced cysteine residues and the samples were incubated for 30 min in darkness. The filters were washed with 100 μl UA buffer three times and then 100 μl 100 mM TEAB buffer twice. Finally, the protein suspensions were digested with 4 μg trypsin in 40 μl TEAB buffer overnight at 37 °C, and the resulting peptides were collected as a filtrate. The peptide content was estimated by UV light spectral density at 280 nM using an extinctions coefficient of 1.1 of 0.1% (g/l) solution that was calculated on the basis of the frequency of tryptophan and tyrosine in vertebrate proteins.

Following this and according to the manufacturer’s instructions (Thermo Fisher Scientific, Waltham, MA, USA), a peptide mixture of each sample (100 μg) was labeled with the TMT reagent.

Peptide fractionation with high pH reversed phase

According to the manufacturer’s instructions, increasing the acetonitrile step-gradient elution, the TMT-labeled digest samples into 10 fractions fractionated with a Pierce high pH reversed-phase fractionation kit (Thermo Fisher Scientific, Waltham, MA, USA).

Mass spectrometry

Each fraction was injected in order to analyze the nano-LC-MS/MS. A reverse-phase trap column which is connected to a C18 reversed-phase analytical column in buffer A (0.1% formic acid) and separated with a linear gradient of buffer B (84% acetonitrile and 0.1% formic acid) loaded the peptide mixture and the flow rate was 300 nL/min and it was controlled by IntelliFlow technology.

The LC-MS/MS analysis was completed with a Q Exactive mass spectrometer and it was coupled to Easy nLC for 60 min. The mass spectrometer was performed in positive ion mode. Using a data-dependent top10 method can obtain the MS data, and the most abundant precursor ions were selected from the scan of 300–1,800 m/z. An automatic gain control (AGC) target was 3e6, and the maximum inject time was 10 ms. Dynamic exclusion duration was 40 s. Survey scans were obtained with the resolution of 70,000 at m/z 200 and the resolution for the HCD spectra was 17,500 with m/z 200, 35,000 at m/z 200, and the width of isolation was 2 m/z. The normalized collision energy was 30 and the under fill ratio specified the minimum percentage of the target value was defined as 0.1%. The instrument was performed by using peptide recognition mode enabled. The LC–MS/MS spectra were searched by using a MASCOT engine which was embedded into Proteome Discoverer 1.4. The MS proteomics data have been deposited to the ProteomeXchange Consortium via the PRIDE (Perez-Riverol et al., 2019) partner repository with the dataset identifier PXD017804.

PRM analysis

We further performed LC-PRM analysis (Peterson et al., 2012) for several selected proteins from the same samples, to test the protein abundance obtained by previous TMT analysis. We prepared the peptides and then analyzed by using TMT analysis according to the protocol, and TOMHAQ has used for validation (Shanghai Applied Protein Technology Co., Ltd., Shanghai, China) (Zhang et al., 2019). The raw data were analyzed with the bioinformatic tool Skyline (MacLean et al., 2010) (MacCoss Lab, University of Washington, Seattle, WA, USA), where the signal intensities of the peptide sequences for the significantly altered proteins were quantified relative to each sample, and normalized to the standard reference (Liu & Lv, 2019).

Statistical and bioinformatics analysis

Basic information including participants’ age, age at menopause, duration of menopause, BMI, biochemical detection, BMD and T score were displayed as the mean ± standard deviation (SD). A P-value < 0.05 with two-tailed Student’s t-test was considered statistically different between two groups. A correlation analysis between basic information, protein expression, and BMD values was performed using R 3.6.0 tools and GraphPad Prism 8.0.1 statistical software (GraphPad, La Jolla, CA, USA). Proteins with a fold change (FC) > 1.2 as well as a statistical P-value < 0.05 between two groups were selected as DAPs. The unsupervised principal component analysis (PCA) and partial least squares discriminant analysis (PLS-DA) model of proteome data were used in this study. PCA and PLS-DA analysis of proteome data, and the bi-clustering and volcano plots of DAPs were performed with R 3.6.0 tools. We searched Blast2GO (version 3.3.5) during the gene ontology mapping and annotation procedure (Gotz et al., 2008). GO enrichment on three modules including biological process (GO-BP), molecular function (GO-MF), and cellular component (GO-CC) was applied based on a Fisher’ exact test, with P-values < 0.05 considered as statistically significant functional categories.

Results

Characteristics of the study population and correlation analysis

Our study recruited a total of 54 postmenopausal females, including 18 females with normal bone mass (normal controls (NC)), 18 females with ON and 18 with OP. We collected and analyzed patients’ information, including their age, age at menopause (M_age), body mass index (BMI), biochemical detection results and BMD measurement results (BMD value and T score data from the DXA–BMD detection). Figure 2 displays the DXA scan images obtained during the BMD measurement using the detector. As for BMD and T scores, when comparing the ON vs. NC, OP vs. NC, and OP vs. ON groups, all showed statistically significant differences, which is consistent with the diagnostic criteria for OP and ON. There were no statistically significant differences between any of the two groups in terms of age, age at menopause, and serum concentration of Vitamin, ALP, P1NP and CTX, while the participants in OP group showed a significant lower BMI compared with that in NC and ON groups (Table 1).

Table 1 Participant characteristics.

Group	n	Age	M-age	BMI	Vitamin
(ng/mL)	ALP
(U/L)	P1NP
(ng/mL)	CTX
(ng/mL)	BMD
(g/cm2)	T score	
Normal control	18	55.22 ± 5.31	48.06 ± 2.70	25.02 ± 2.86	17.81 ± 4.36	99.33 ± 28.85	77.90 ± 26.55	0.44 ± 0.15	0.92 ± 0.11	–0.47 ± 0.57	
Osteopenia	18	56.72 ± 4.92	47.11 ± 1.71	24.71 ± 2.58#	19.05 ± 5.54	108.39 ± 37.47	70.98 ± 26.49	0.41 ± 0.14	0.81 ± 0.06#	–1.94 ± 0.39#	
Osteoporosis	18	58.33 ± 5.40	46.83 ± 2.07	21.89 ± 1.64#Δ	19.17 ± 7.27	109.61 ± 27.86	90.93 ± 42.68	0.49 ± 0.18	0.61 ± 0.24#Δ	–4.50 ± 0.26#Δ	
Total	54	56.76 ± 5.27	47.32 ± 2.21	23.87 ± 2.76	18.69 ± 5.77	105.78 ± 31.43	79.94 ± 33.25	0.45 ± 0.16	0.78 ± 0.20	–2.31 ± 1.73	
Notes:

# Compared with the NC group, P < 0.01.

Δ Compared with the ON group, P < 0.01.

M-age, age at menopause; BMI, body mass index; BMD, bone mineral density; Vitamin, 25-hydroxyvitamin D; ALP, alkaline phosphatase; P1NP, procollagen type 1 N-peptide; CTX, C-terminal cross-linking telopeptide of type 1 collagen.

We further performed Pearson’s correlation analysis between the different indicators to explore BMD-related factors. Overall, the BMD values were negatively correlated with age (Fig. 3A). Meanwhile, a negative correlation was also observed between the menopause duration and BMD values (Fig. 3C), while a positive correlation was found between BMD and age at menopause (Fig. 3B), and between BMD and BMI (Fig. 3D). Besides, we observed a positive correlation between the concentration of Vitamin D and BMD values, although not statistically significant (Fig. S1A), and the concentration of ALP, P1NP, CTX was not significantly related with the BMD values (Figs. S1B–S1D).

Figure 3 Correlation analysis of age, age at menopause, duration of menopause, BMI and BMD.

(A) Age was negatively related with BMD values. (B) Age at menopause was positively related with BMD values. (C) Duration of menopause was negatively related with BMD values. (D) BMI was positively related with BMD values.

DAP analysis among OP, ON, and NC with proteomic data

Before the proteomics experiment, we randomly mixed six samples into one sample in each group; that is, three mixed plasma samples in each group were prepared for further proteome analysis. High-throughput, label-free, liquid chromatography–tandem mass spectrometry (LC–MS/MS) experiments identified a total of 1,092 proteins in the plasma samples in our study. The unsupervised principal component analysis (PCA) and partial least squares discriminant analysis (PLS-DA) model of proteome data revealed evidence of the separation between low BMD (OP and ON) and the NC, especially when discriminating between OP and NC samples, while it showed less-than-ideal effects on the separation between the OP and ON samples (Figs. 4A and 4B). We selected the proteins with a FC > 1.2 (i.e., where upregulation was >1.2 times or downregulation <0.83 times), and P < 0.05 indicated a statistically significant difference in the DAPs between two compared groups. When compared with the NC group, there were 17 upregulated and 20 downregulated DAPs in the ON group (Figs. 4C and 4D), and 32 upregulated and 123 downregulated DAPs in the OP group (Figs. 4C and 4E). Compared with the ON group, there were 11 upregulated and 5 downregulated DAPs in the OP group (Figs. 4C and 4F). If we set the FC > 1.5 (i.e., when the upregulation was >1.5 times or downregulation <0.67 times), P < 0.5 as the criteria for DAPs, there were 8, 127 and 3 DAPs in the ON vs. NC, OP vs. NC and OP vs. ON groups, respectively (Fig. 4C). The details of the protein quantification results and differential analysis between each group comparison are presented in Supplemental Files 1 (ON vs. NC), Supplemental Files 2 (OP vs. NC) and Supplemental Files 3 (OP vs. ON). The peptide lists of all the identified proteins in the study are presented in Supplemental Files 4.

Figure 4 PCA and PLS-DA analysis of protein expression and DAPs between each of the groups.

(A) PCA score plots of proteome data. (B) PLS-DA score plots of proteome data. (C) Number of DAPs between each group. (D) Volcano plot of DAPs between ON vs. NC when FC > 1.2. (E) Volcano plot of DAPs between OP vs. NC when FC > 1.2. (F) Volcano plot of DAPs between OP vs. ON when FC > 1.2.

These results may suggest that the proteome of OP samples were different for the NC group, whereas the ON group (the middle group) showed greater similarity with the OP group, as there was a partly overlapping PLS-DA model between the OP and ON groups and the number of DAPs was less in the OP vs. ON group when compared with the ON vs. NC group. Thus, in subsequent analyses, we placed greater focus on the difference between low BMD (OP and ON) and NC. Functional analysis indicated that the DAPs in the ON vs. NC groups mainly demonstrated catalytic, binding, and molecular function regulator activity; the DAPs appeared to be involved in metabolic, cellular, stimulus response, biological regulation and immune system processes (Fig. 5A). Similarly, the DAPs in the OP vs. NC groups mostly exhibited binding, molecular function regulator, transporter, and molecular transducer activity, and appeared to be involved in cellular, stimulus response, biological regulation, metabolic and biological regulation processes (Fig. 5B).

Figure 5 Functional analysis of DAPs between the ON vs. NC and OP vs. NC groups.

(A) Biological process, molecular function, and cellular component annotation of DAPs in the ON vs. NC groups. (B) Biological process, molecular function and cellular component annotation of DAPs in the OP vs. NC groups.

Identification of candidate proteins correlated with low BMD values in PMOP

In our study, most of the DAPs existed in the OP vs. NC groups, followed by the ON vs. NC and OP vs. ON groups. We focused on the DAPs in the low BMD group, which was found in both the OP vs. NC and ON vs. NC comparisons. When compared with the normal BMD samples, seven proteins were differentially expressed in the OP and ON groups, including Adipocyte plasma membrane-associated protein (H0Y512), Protein disulfideisomerase A5 (Q14554), cDNA FLJ51711 (B4DE30), Glucosidase (A0A024R592), Lysozyme C (P61626), Uncharacterized protein DKFZp666N164 (Q658S4) and Complement C4-A (P0C0L4) (Table 2). Figure S2 showed the MS/MS spectra for the seven DAPs in the low BMD groups. Bi-clustering analysis of these seven DAPs indicated a clear discrimination between normal BMD and low BMD samples (Fig. 6A). The detailed expression of the seven proteins in each group were showed in Figs. 6B–6H. The expression of protein protein Lysozyme C was positively related with BMD values (Fig. S3A), while the expression of proteins proteins Glucosidase, cDNA FLJ51711, and Protein disulfideisomerase A5 was negatively related with BMD values (Figs. S3B–S3D). The correlation between the expression of Complement C4-A, Uncharacterized protein DKFZp666N164, Adipocyte plasma membrane-associated protein, and BMD values was not highly significant (Figs. S3E–S3G). The reported numbers of proteins were supported with more than one peptide.

Table 2 Changes in the plasma proteins between low BMD and normal controls.

Accession	Protein name (description)	Average NC	Average ON	Average OP	ON vs. NC	OP vs. NC	
FC	P	FC	P	
P61626	Lysozyme C	1.249	0.949	0.820	0.760	0.019	0.657	0.003	
A0A024R592	Glucosidase	0.880	1.118	1.091	1.270	0.033	1.239	0.035	
B4DE30	cDNA FLJ51711	0.813	1.073	1.172	1.320	0.041	1.442	0.037	
Q14554	Protein disulfide isomerase A5	0.697	1.098	1.217	1.575	0.037	1.746	0.040	
P0C0L4	Complement C4-A	1.516	0.771	0.764	0.508	0.023	0.504	0.020	
Q658S4	Uncharacterized protein DKFZp666N164	1.350	0.910	0.855	0.674	0.040	0.633	0.031	
H0Y512	Adipocyte plasma membrane-associated protein	0.719	1.225	1.013	1.704	0.015	1.410	0.014	

Figure 6 Bi-clustering of the seven DAPs and correlation analysis of DAPs with BMD values.

(A) Bi-clustering analysis of seven DAPs in normal BMD and low BMD samples. (B–H) The detailed expression of the seven proteins in each group. P61626: Lysozyme C; A5A0A024R592: Glucosidase; B4DE30: cDNA FLJ51711; Q14554: Protein disulfideisomerase A5; P0C0L4: Complement C4-A; Q658S4: Uncharacterized protein DKFZp666N164; H0Y512: Adipocyte plasma membrane-associated proteinproteins.

Verification of the candidate protein biomarkers associated with low BMD with PRM

In this study, Adipocyte plasma membrane-associated protein, Protein disulfideisomerase A5, cDNA FLJ51711, Glucosidase, Lysozyme C, Uncharacterized protein DKFZp666N164 and Complement C4-Awere differentially expressed in OP and ON group compared with normal sample. However the correlation between the expression of Complement C4-A, Uncharacterized protein DKFZp666N164, Adipocyte plasma membrane-associated protein, and BMD values was not highly significant. Therefore we further selected the four BMD-related DAPs including Lysozyme C, Glucosidase, cDNA FLJ51711, and Protein disulfideisomerase A5 for validation using the PRM approach. Among them, the abundance of Lysozyme C was also decreased in the low BMD groups, while Glucosidase and Protein disulfideisomerase A5 expression were increased as BMD decreased (Fig. 7). These results from the PRM testing suggested that the candidate proteins demonstrated similar trends as the TMT results, which confirmed the credibility of the proteomics data and verified the DAPs between the NC and low BMD samples.

Figure 7 Expression patterns of selected DAPs using TMT analysis and PRM validation.

(A) Abundance of protein Lysozyme C (P61626). (B) Abundance of protein Glucosidase (A5A0A024R592). (C) Abundance of protein cDNA FLJ51711 (B4DE30). (D) Abundance of protein Protein disulfideisomerase (Q14554). *P < 0.05 between two groups; **P < 0.01 between two groups.

Discussion

The risk of developing OP and broken bones is increasing in association with advanced age. Several factors have contributed to the result that women are more likely to develop OP than men. First, women tend to have smaller, thinner bones than men (Cooper et al., 2007). Second, estrogen levels decline dramatically when women reach menopause, which results in consequent bone loss (Ikeda, Horie-Inoue & Inoue, 2019). This explains why the chance of developing OP increases once a woman reaches menopause and why that risk increases with age.

During the process of bone remodeling, biochemical products are produced and can be detected in the blood and/or urine, which are considered to be biomarkers. Plasma and serum are rich sources of information regarding an individual’s health state, and protein tests inform medical decision making (Geyer et al., 2019; Lin et al., 2018). Biomedical substances that are involved in metabolic activities are tied to an individual’s biological or metabolic status; thus, circulating proteins are considered sensitive and specific markers of certain pathological states, including abnormal bone metabolism (Brosseron et al., 2018; Shao et al., 2019). Thus far, no studies have been reported that directly focused on the plasma proteome in Chinese PMOP females. There are several studies that have analyzed the quantitative proteome profiles and explored particular BMD-related proteins or genes based on peripheral blood monocytes (Shao et al., 2019; Zhang et al., 2016). Recently, a comparative proteomics analysis of serum microvesicles for the evaluation of OP was reported, and the authors found that Profilin 1 could differentiate those in the OP group from those in the ON and normal groups (Huo, Li & Qiao, 2019).

Our study is the first to investigate plasma proteomic profiling in Chinese postmenopausal women with OP, ON, and normal bone mass using TMT and PRM approaches. Overall, by means of the PCA and PLS-DA model, a relative separation between the low BMD and NC groups was found, and the proteome expression patterns in the ON samples demonstrated greater similarity with the OP samples, as compared with the NC samples, which indicates that the alteration of an individual’s plasma proteome profile is sensitive and specific to BMD changes, thus making it possible to discriminate between postmenopausal women with low BMD from those with normal bone mass. The number of DAPs between each of the two groups also confirmed the above conclusion, that OP vs. NC had the most numerous DAPs, followed by ON vs. NC and OP vs. ON (Figs. 4C–4F). Combing the TMT proteome data and PRM validation testing, we finally identified three candidate proteins that correlated with BMD values (Figs. 6 and 7). This provides the basis for identifying novel biomarkers and exploring the pathogenesis related to abnormal bone metabolism in the field of PMOP research. In addition, there were no statistically significant differences between any of the two groups in terms of age and age at menopause, while the participants in OP group showed a significant lower BMI compared with that in NC and ON groups (Table 1). Therefore we conclude that the proteins observed were not due to differences in the BMI between the groups studied.

The specific effects of protein lysozyme C, glucosidase and protein disulfideisomerase A5 on OP are still unclear. Studies have reported the correlation with low BMD/OP and the late-onset Pompe disease (LOPD) (Papadimas et al., 2011; Sheng et al., 2017). LOPD is a lysosomal storage disease resulted from deficiency of the enzyme acid α-glucosidase, and enzyme replacement therapy (ERT) with alglucosidase alfa is the only specific treatment available (Van der Ploeg et al., 2010). In a recent research, patients with LOPD showed improvement in BMD after alglucosidase ERT (Sheng et al., 2017). Protein disulfideisomerase A5 belongs to the protein disulfideisomerase (PDI) family. Studies demonstrated that some murine models for OP is related to endoplasmic reticulum (ER) stress response of osteoblasts (Hino et al., 2010; Li et al., 2018). The ER molecular chaperones including PDI are down-regulated in osteoblasts from OP patients (Hino et al., 2010). Icariin (ICA, a flavonol glycoside isolated from a traditional Chinese medicinal herb Epimedium sagittatum) have demonstrated its anti-osteoporotic and osteogenic differentiation effects (Chen et al., 2005; Meng et al., 2005). Recently, secretome analysis demonstrated an upregulation of the expression of protein disulfideisomerase family A, member 3 (PDIA3) after ICA treatmentinduced osteogenesis in rat osteoblasts (Qian et al., 2018). Considering the above reports and the results from our study together, glucosidase and protein disulfideisomerase may be potential targets of therapeutic agents for the prevention of bone loss in the low BMD patients.

In this study, we identified differentially expressed proteins in OP and ON samples when compared to NCs. Additionally, a selected subgroup of differentially expressed proteins was further validated using the PRM approach. However, it’s restricted to the number of clinical sample and follows up study, what we identified in this study are very primary. And further validation for our findings need to be completed both in lab and in clinic. And the specificity and sensitivity of candidate biomarkers we found also need to be improved.

Conclusions

In conclusion, the study presented a proteome profiling analysis using TMT and PRM approaches obtained from postmenopausal females with OP, ON, and normal bone mass. The PLS-DA model of the protein expression data clearly distinguished OP from the normal samples. We also identified the differentiating abundant proteins, some of which showed significant correlations with BMD values. These data suggest that these protein profiles are indicators of bone metabolism status; the candidate proteins are associated with decreasing BMD values and may presumably represent an increased risk of fracture. These markers may provide information that guides the monitoring of both disease progression and treatment efficacy for PMOP females.

Supplemental Information

Supplemental Information 1 Raw data of the basic information of all the participants.

Click here for additional data file.

Supplemental Information 2 Protein quantification and differential analysis in samples between ON and NC groups.

The red filled columes represent up-regulation DAPs, and green filled columes represent down-regulation DAPs

Click here for additional data file.

Supplemental Information 3 Protein quantification and differential analysis in samples between OP and NC groups.

The red filled columes represent up-regulation DAPs, and green filled columes represent down-regulation DAPs

Click here for additional data file.

Supplemental Information 4 Protein quantification and differential analysis in samples between OP and ON groups.

The red filled columes represent up-regulation DAPs, and green filled columes represent down-regulation DAPs

Click here for additional data file.

Supplemental Information 5 The peptide lists of all the identified proteins in the study.

Click here for additional data file.

Supplemental Information 6 Correlation analysis of biochemical detection and BMD values.

Click here for additional data file.

Supplemental Information 7 The MS/MS spectra for the seven DAPs.

Click here for additional data file.

Supplemental Information 8 Correlation analysis of DAPs with BMD values.

(A) The abundance of protein Lysozyme C (P61626) was positively related with BMD values. (B–D) The abundance of proteins Glucosidase (A5A0A024R592), cDNA FLJ51711 (B4DE30), and Protein disulfideisomerase (Q14554) was negatively related to BMD values. (E-G) The abundance of proteins Complement C4-A (P0C0L4), Uncharacterized protein DKFZp666N164 (Q658S4), and Adipocyte plasma membrane-associated proteinproteins (H0Y512) was not significantly related to BMD values.

Click here for additional data file.

The authors are grateful to the Shanghai Applied Protein Technology (Shanghai, China) for providing technical assistance with data analysis.

Additional Information and Declarations

Competing Interests

Author Contributions

Human Ethics

Data Availability

The authors declare that they have no competing interests.

Dageng Huang conceived and designed the experiments, prepared figures and/or tables, and approved the final draft.

Yangyang Wang analyzed the data, prepared figures and/or tables, and approved the final draft.

Jing Lv performed the experiments, prepared figures and/or tables, and approved the final draft.

Yuzhu Yan performed the experiments, authored or reviewed drafts of the paper, and approved the final draft.

Ya Hu performed the experiments, authored or reviewed drafts of the paper, and approved the final draft.

Cuicui Liu performed the experiments, authored or reviewed drafts of the paper, and approved the final draft.

Feng Zhang analyzed the data, prepared figures and/or tables, and approved the final draft.

Jihan Wang conceived and designed the experiments, prepared figures and/or tables, authored or reviewed drafts of the paper, and approved the final draft.

Dingjun Hao conceived and designed the experiments, prepared figures and/or tables, authored or reviewed drafts of the paper, and approved the final draft.

The following information was supplied relating to ethical approvals (i.e., approving body and any reference numbers):

The project was approved by the Institutional Review Board, Honghui Hospital, Xi’an Jiaotong University (2018-22).

The following information was supplied regarding data availability:

The data are available in the Supplemental Files. The details of the protein quantification results and differential analysis between each group comparison are presented in Supplemental File 1 (ON vs. NC), Supplemental File 2 (OP vs. NC) and Supplemental File 3 (OP vs. ON). The peptide lists of all the identified proteins in the study are presented in Supplemental File 4.

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
