# Peer review of "Proteomic profiling analysis of postmenopausal osteoporosis and osteopenia identifies potential proteins associated with low bone mineral density"

_PeerJ, doi:10.7717/peerj.9009_

## Round 0.1 · original submission · Major Revisions

Please address critiques of all reviewers and amend your manuscript accordingly

·

Basic reporting

The methodology part is missing

Experimental design

No Comment

Validity of the findings

No Comment

Additional comments

This manuscript by Huang et al. describes proteomic profiling of the plasma from postmenopausal women with or without low BMD (osteoporosis and osteopenia) using a tandem mass tag (TMT) labeling followed by the validation of the candidates using parallel reaction monitoring. This study highlight important aspect of the Postmenopausal osteoporosis and will be accepted after following major concern have been addressed

1. The mass spectrometry data needs to be deposited in the public repository
2. The method section were not written properly and it was not possible to reproduce the results using the method. For e.g. The authors used Agilent System Column for the depletion. The catalog number column details are missing
3. The parallel reaction monitoring assay used for validation. The method section was incomplete. It is written that TMT labelled peptide were used for PRM analysis. This means author has used TOMHAQ for validation. If not mentioned same in the method section
4. The author has mention normalized collision energy in ev. NCE does not have unit.
5. Figure 1 does not mentioned about depletion
6. The author should elaborate on the rational for the shortlisting candidates for the validation
7. The mass spectrometry data analysis information is completely missing. How this differentially regulated molecules identified
8. The reported number of proteins were supported with more than 1 peptides. Add this information in the results
9. Why the samples were pooled for the TMT experiment. It can be done individually either with 11 plex or 16 plex TMT to get sample specific information

Reviewer 2 ·

Basic reporting

Your introduction needs more detail. I suggest that you improve the description at lines 64-71. The authors should contextualize the study better and acknowledge other studies about proteome serum and bone mineral density. I suggest to add references related to these issues as:
1. Bhattacharyya, S.; Siegel, E.R.; Achenbach, S.J.; Khosla, S.; Suva, L.J. Serum biomarker profile associated with high bone turnover and BMD in postmenopausal women. Journal of Bone and Mineral Research , 2008, 23, 1106–1117.

2: Martínez-Aguilar MM, Aparicio-Bautista DI, Ramírez-Salazar EG, et al. Serum Proteomic Analysis Reveals Vitamin D-Binding Protein (VDBP) as a Potential Biomarker for Low Bone Mineral Density in Mexican Postmenopausal Women. Nutrients. 2019 Nov 21;11(12).

Experimental design

The Authors should better describe the eligibility criteria they used to enroll the subjects participating the study. In particular they should clarify if they excluded or not (and why) subjects with metabolic diseases or taking drugs influencing the bone metabolism. If not, how this could influence the results of the study.

I suggest paying special interest to the BMI. The authors, as they ensure that the proteins observed, were not due to differences in the BMI between the groups studied, this point should be discussed in detail.

The data of biochemical measurements should be reflected in Table 1 Participant characteristics or Supplementary Basic_information.

A major cause for concern is the statistical approach, and validity of a t-test when only n=2 replicates are analyzed for the three conditions after sample pooling. Furthermore, given that multiple variables are tested, what adjustment is made for multiple comparisons?

Validity of the findings

In the discussion, I would spend some word comparing the performance of these proteins in respect to other candidate biomarkers, on how to combine, in the future, with other markers to improve specificity and sensitivity.

The authors should discuss in detail the limitations and strengths of their study to enrich the manuscript.

Perhaps, the most important limitation of the present paper is the lack of follow-up. This issue should be further clarified.

Another point that should be clarified, is because they did not individually validate the total of the samples, n = 54.

Additional comments

Firstly, I would like to thank the journal for letting me review such an interesting paper. The look for new biomarkers is a field of great interest. The present paper has several strengths, such as the proteomic approach. However, in my opinion, several drawbacks should be clarified.

Data quality is acceptable overall. However these results could be regarded as preliminary and limited, mainly due to the methodology used, and do not significantly advance the cause of biomarker development for OS/OP.

Reviewer 3 ·

Basic reporting

1. The authors don’t mention if they have submitted their proteomic data to a public proteomic repository. The authors should submit their proteomic data to some public repository or database where the data files can be accessed by other researchers.

Experimental design

1. The authors also need to mention the product code or catalog information for the Multiple Affinity Removal Column Human 14 or some other Agilent serum depletion columns they have used in the study. So that other researchers can in future can replicate their findings using the same columns

2. The authors also don’t mention anything about the database they have used to search their TMT-labeled MS data against. It looks like they have used UniProt database. The authors should provide information regarding the version of the database they have used in the study and the number of the protein entries in the database

3. The authors need to describe in detail the in-solution peptide digestion protocol they used to generate the peptides which were further labeled with TMT-tags.

Validity of the findings

Major points:

1. In the introduction section line 43-44: the author talk of global prevalence of OP cases in future (2035- 48.03 million cases; 2050- 5.99 million cases) according to Shi et al. 2015. However, the numbers provided by authors need to be corrected as the real numbers quoted by Shi et al is as follows
“Projection of annual fractures and related cost to 2050: Projection of fractures by sex for each fracture site is given in Fig. 3 and Appendix Table 3. Fracture number and related costs at the included fracture sites were estimated to increase through to 2050 in both sexes. Relative to the base year, annual total fracture number and costs were predicted to double by year 2035 (4.83 million fractures at a cost of 19.92 billion US dollars) and were projected to rise to 5.99 (95 % CI 5.44, 6.56) million fractures, accounted for approximately 9.84 per 1000 people aged 50+ years, costing $25.43 billion (95 % CI 23.92, 26.95 billion US dollars) by year 2050.”
2. The authors have strangely not provided the peptide list of all the identified proteins in the study for OP, ON and CN. Especially, the differentially abundant proteins (DAPs). This call into question the authenticity of the data
3. One major criticism of the study is the authors provide no MS/MS spectra for the DAPs at least MS/MS spectra for one peptide for each of the eight DAPs mentioned in Table 2

Minor points:
1. The authors should abbreviated the term SDT buffer on line 113 in the methods section of the manuscript
2. The authors need to name the proteins mentioned on line number 234 in the result section of the manuscript. It is highly inconvenient for the readers to check table 2 each time.
3. In the abstract of the manuscript line 28-29 the authors need to rephrase the sentence to “Collectively, our results suggests that postmenopausal women with low BMD have a different proteomic profile or signature”

Additional comments

The manuscript by Huang et al. utilized Tandem Mass Tag (TMT) labeling-based quantitative proteomic approach along with parallel reaction monitoring (PRM) approach to study the comparative serum proteome from postmenopausal osteoporosis (OP), osteopenia (ON) and age matched healthy controls (CN). The authors identified differentially expressed proteins in serum of OP and ON samples when compared to controls (CN). Additionally, a selected subgroup of differentially expressed proteins was further validated using the PRM approach. I strongly feel that the work undertaken by the authors is quite important in the field of postmenopausal osteoporosis considering the lack of osteoporosis associated biomarkers and early detection disease markers. However, the manuscript has major issues and the authors have not provided all the experimental and result information in the current version of the manuscript. The authors have to address the issues before the manuscript could be accepted for publication.

---

## Round 0.2 · accepted · Accept

All critiques were addressed and the manuscript was revised accordingly. Therefore, the amended version is acceptable.

Reviewer 2 ·

Basic reporting

no comment

Experimental design

no comment

Validity of the findings

no comment

Additional comments

no comments

Reviewer 3 ·

Basic reporting

The authors have successfully addressed most of the queries in the revised version of the manuscript.

Experimental design

The authors have successfully addressed most of the queries in the revised version of the manuscript.

Validity of the findings

The authors have successfully addressed most of the queries in the revised version of the manuscript.